# The Course of COVID-19 among Unvaccinated Patients—Data from the National Hospital in Warsaw, Poland

**DOI:** 10.3390/vaccines11030675

**Published:** 2023-03-16

**Authors:** Artur Zaczyński, Michał Hampel, Paweł Piątkiewicz, Jacek Nasiłowski, Sławomir Butkiewicz, Urszula Religioni, Agnieszka Barańska, Jolanta Herda, Agnieszka Neumann-Podczaska, Regis Vaillancourt, Piotr Merks

**Affiliations:** 1Central Clinical Hospital of the Ministry of Interior and Administration in Warsaw, 02-507 Warsaw, Poland; 2Department of Pharmacology and Clinical Pharmacology, Faculty of Medicine, Collegium Medicum Cardinal Stefan Wyszyński, 01-815 Warsaw, Poland; 3Department of Internal Medicine, Pulmonary Diseases and Allergy, Medical University of Warsaw, 02-091 Warsaw, Poland; 4School of Public Health, Centre of Postgraduate Medical Education of Warsaw, 01-826 Warsaw, Poland; 5Department of Medical Informatics and Statistics with e-Health Lab, Medical University of Lublin, 20-124 Lublin, Poland; 6Chair and Department of Palliative Medicine, Poznan University of Medical Sciences, 61-245 Poznan, Poland

**Keywords:** COVID-19, unvaccinated patients, course of COVID-19, clinical evidence, Poland

## Abstract

Introduction. Studies to date indicate the relatively high effectiveness of vaccinations in preventing severe COVID-19 symptoms. However, in Poland, 40% of the population remains unvaccinated. Objective. The objective of this study was to describe the natural history of COVID-19 in unvaccinated hospital patients in Warsaw, Poland. Material and methods. This study evaluated data from 50 adult patients from the National Hospital in Warsaw, Poland, in the period 26 November 2021 to 11 March 2022. None of these patients had been vaccinated against COVID-19. Results. Analysis showed that the average hospitalisation time for these unvaccinated COVID-19 patients was 13 days. Clinical deterioration was observed in 70% of these patients, 40% required the intensive care unit, and 34% subsequently died prior to the end of the study. Conclusions. There was a significant deterioration and high mortality rate in the unvaccinated patients. For this reason, it seems prudent to take measures to increase the vaccination coverage level of the population against COVID-19.

## 1. Introduction

The COVID-19 pandemic has been one of the largest global health issues. As of November 2022, around 632 million people have been infected with COVID-19, and 6.6 million have died [1]. In Poland, 6.34 million confirmed cases of COVID-19 and 118 thousand deaths have been reported (as of 4 November 2022). Globally, 68.2% of the population has received at least one dose of a vaccine against COVID-19, and in Poland, this figure is 59.8% [1]. Currently, there are several vaccines against COVID-19 approved in the European Union: Pfizer/BioNTech Comirnaty (mRNA) (BNT162b2) for those ≥5 years of age; updated booster vaccines against Omicron, Comirnaty Original/Omicron BA.1. (Pfizer-BioNtech), Comirnaty Original/Omicron BA.4/BA.5. (Pfizer-BioNtech); Moderna Spikevax (mRNA 1273) for those ≥6 years of age; updated booster vaccines against Omicron, Spikevax Bivalent Original/Omicron BA.1. (Moderna); AstraZeneca vector Vaxzevria (ChAdOx1 nCoV-19) vaccine for those ≥18 years of age in two doses; Jcovden (COVID-19 Vaccine Janssen) (Janssen) (Ad26.COV.2-S) for those ≥18 years of age; and Novavax protein-based Nuvaxovid (NVX-CoV2373) vaccine for those ≥18 years of age [2]. Previous studies have shown the relatively high effectiveness of these vaccines in preventing a severe course of infection, hospitalisation, and death due to COVID-19 [3,4,5]. The length of protection, the necessity and schemes of boosters, the dependencies of protection on the demographic characteristics of the patient, and the changing variants of the coronavirus are still under discussion [5,6,7,8].

Despite the high effectiveness of the vaccines in preventing a severe course of the disease, a large proportion of the global population has not been vaccinated at all, particularly in low-income countries, where only 23.4% of people have received one vaccine dose or more. Although 75% of European Union citizens have been vaccinated with at least one vaccine dose [1], in Poland, about 40% of the population remain unvaccinated (as of January 2023), making Poland one of the least vaccinated countries in the European Union. Studies carried out in Poland show that about half of society is still uncertain about the safety and effectiveness of vaccines [9,10]. 

Despite the many publications on the course of COVID-19 in vaccinated hospital patients, it is difficult to find studies describing the course of COVID-19 solely in unvaccinated hospital patients. In particular, this type of data is missing for Polish patients. The current study takes into account patients hospitalised at the main COVID hospital in Poland, i.e., the National Hospital.

Considering the above, the primary objective of this study was to describe the course of COVID-19 in unvaccinated hospital patients.

## 2. Materials and Methods

Data were collected from the National Hospital in Warsaw, Poland, from 26 November 2021 to 11 March 2022 using the Clininet system (https://www.cgm.com/_Resources/Persistent/f7d7adc241aebacc931c3db67f614e963df90dd4/2019-09-09%20CGM%20CLININET%20PL.pdf, accessed on 30 August 2022.).

The National Hospital was a temporary COVID-19 hospital set up at the National Stadium in Warsaw in October 2020 by the Central Clinical Hospital of the Ministry of Interior and Administration (CSK MSWiA). 

Each patient admitted to the Provisional Hospital at the National Stadium (STSN) could be admitted in three ways:(1)Transferred directly by the emergency medical team;(2)Transferred from another hospital after prior qualification by the coordination centre Temporary Hospital, after the patient was reported by the hospital;(3)Transferred from the emergency department or departments of the home centre—i.e., the Central Hospital.

The Clinical Hospital of the Ministry of Interior and Administration in Warsaw served as a specialised single-name hospital. Each patient admitted to ST SN was qualified for admission on the basis of criteria developed by the therapeutic committee of the Central Clinical Hospital of the Ministry of Interior and Administration. This evaluation was carried out by the staff of the ST Coordination Centre, including the duty officer, a trained paramedic, and an experienced doctor working in the OR.

As part of the Temporary Hospital, there was an emergency room that carried out patient movement admission/discharge. During admission to the hospital in the Admission Room of ST SN, the patient was triaged in order to confirm the need for hospitalisation by the Coordination Centre; then, basic diagnostic and laboratory tests were performed.

If it was found necessary to extend diagnostics, e.g., with imaging tests such as a CT, ultrasound, or other examination, this examination was performed and ordered from the ward level parties.

During the admission, the patient was informed about their state of health and where they would be hospitalised, and signed informed consent for hospitalisation. Patient records were kept in paper and electronic form.

The triage study was supplemented by the CliniNet hospital system coupled to the system of the parent centre, i.e., CSK MSWiA. All other examinations, and medical and nursing observations were supplemented in the hospital system.

A reassessment of the patient’s condition was performed in order to select the appropriate “segment” in the temporary hospital to which the patient should be transferred from the emergency room parties. In the temporary hospital, there were “sections” according to the degree of required care. Standard sections where treatment was carried out distinguished between mild to moderate patients. Patients with severe respiratory failure, requiring high-flow oxygen therapy, were in the “pre-ICU” section and patients on ventilator therapy were in a serious condition in the intensive care section.

The pre-ICU and ICU staff consisted of anaesthesiologists and intensive care physicians, as well as nurses specialising in anaesthesiology and intensive care.

Vital parameters were measured three times a day and recorded in the hospital system. Data on the patient’s stay were archived in paper and electronic forms; thus, the data were obtained from the hospital system and paper documentation.

The analysis included 50 patients hospitalised during the study period who had not previously been vaccinated against COVID-19.

## 3. Results 

### 3.1. Characteristics of the Study Group

The analysis included 50 unvaccinated patients. The mean age of the patients was 66.82 ± 3.02 years; the youngest was 60, and the oldest was 73 years old. The number of women and men was equal. The BMI of the patients ranged from 21.37 to 40.01, and half of them had a BMI of 27.26 or higher. Demographic characteristics of the patients are presented below in Table 1.

Table 2 lists the comorbidities and risk factors of the studied patients. The main risk factor was nicotine use (8%). The comorbidities included diabetes (26%), autoimmune diseases (10%), atherosclerosis (6%), asthma/COPD (6%), nephropathy (6%), and liver disorders (4%).

The hospitalisation length of the patients infected with COVID-19 ranged from 4 to 104 days, with half of the patients hospitalised for 13 days or longer. Slightly less than half of the patients (40%) needed to stay in the Intensive Care Unit, half of them for 10 days or longer. A total of 34% of the hospitalised patients died. The data on the treatment outcomes are presented below in Table 3.

Table 4 depicts the data on the parameters and conditions on the admission of the patients hospitalised for COVID-19. The average heart rate was 85.54 ± 17.15 bpm, the median systolic blood pressure was 135 ± 25 mmHg, and the average diastolic blood pressure was 80 ± 13 mmHg. The third quartile for temperature was 36.5 °C, which indicates that the vast majority of the patients did not have a high temperature on admission. During O_2_ supplementation, half of the patients had a saturation of 96% or lower and an oxygen partial pressure of 54 mmHg or lower.

On admission, one-third of the COVID-19 patients received oxygen in the form of a nasal cannula (36%) or a non-rebreather mask (34%). Simple oxygen masks and NIV were used less frequently (12% and 4%, respectively), and one in six patients did not require oxygen (14%). Computed tomography revealed lung involvement ranged from 0% to 95%, with at least 30% in half of the patients.

During hospitalisation, the condition of 70% of the patients worsened and required a more invasive form of oxygen therapy, with none of them recording an improvement. A total of 30% of the patients did not require a change in oxygen therapy (Table 5). During hospitalisation, 4% of the COVID-19 patients did not require oxygen therapy, which was significantly less compared to 14% on admission. Nasal cannulas (28% compared to 36%), simple oxygen masks (8% compared to 12%), and non-rebreather masks (16% compared to 34%) were less frequently used, and high-flow oxygen therapy (6% compared to 0%), NIV (non-invasive ventilation i.e., treatment of respiratory failure, which consists of forced airflow in the airways) (8% compared to 4%), and respirator therapy (15% compared to 0%) were more frequent.

Table 6 presents the data on medications administered during hospitalisation. The vast majority of the COVID-19 patients were given dexamethasone (92%) and low-molecule heparin (98%), while baricitinib (20%) and remdesivir (16%) were less common.

### 3.2. The Effects of the Treatments during Hospitalisation

As shown in Table 7 below, statistical analysis revealed that the COVID-19 patients treated with dexamethasone stayed longer in the hospital and the Intensive Care Unit than did the other studied patients. This difference was not statistically significant (*p* = 0.145). Computed tomography recorded a higher percentage of lung involvement, but the difference was not statistically significant (*p* = 0.249). The patients administered dexamethasone more often required a more invasive form of oxygen therapy than those who did not (*p* < 0.001) and were more likely to experience a worsening of their condition (*p* = 0.041). Deaths were higher in the group of patients taking dexamethasone, but this difference was not statistically significant (*p* = 0.692).

Table 8 presents the severity of the COVID-19 infection when taking heparin. No significant differences were observed between the patients who took this medication and those who did not. Due to the fact that only one patient did not take heparin, it was difficult to show statistically significant differences.

The severity of the infection when taking baricitinib is presented in Table 9. The patients who were given baricitinib stayed in the hospital considerably longer than those who were not (*p* = 0.011). The occurrence of deaths and the necessity to stay in the Intensive Care Unit were more frequent in the patients taking this medication. These outcomes were not statistically significant (*p* = 0.654 and *p* = 0.149, respectively). Computed tomography recorded a higher percentage of lung involvement compared to other patients (*p* = 0.026). No statistically significant difference in the stay in the Intensive Care Unit was observed (*p* = 0.629). The patients administered baricitinib were more likely to experience a worsening of their condition, and required more invasive forms of oxygen therapy and ventilation (*p* = 0.021 and *p* = 0.005, respectively). 

As shown in Table 10, the patients who took remdesivir were less likely to experience a worsening of their condition or to require a more invasive form of oxygen therapy (*p* = 0.029), but this difference was not statistically significant (*p* = 0.210). A higher percentage of lung involvement was also found in computed tomography, but this was not statistically significant (*p* = 0.743). 

## 4. Discussion

As far as we know, this conducted analysis of the clinical effects of the course of COVID-19 in unvaccinated hospital patients is the first study of this kind to cover the Polish population. Moreover, the data were collected in the main largest COVID reserve hospital in Poland. Analysis of data on unvaccinated COVID-19 hospital patients shows a significant burden due to the infection. The average length of hospitalisation in this group of patients was 13 days (up to 104 days); 40% of the patients had to stay in the Intensive Care Unit, 70% experienced a worsening of their condition, and 34% subsequently died prior to the end of the study. The course of the infection may have been influenced by the treatment applied. The best outcomes were recorded with remdesivir, yet the studies in this area are not fully conclusive [11,12].

The assessment of mortality among unvaccinated patients carried out by Gullu YT and Koca N indicates that there were more deaths in men, in older patients, and in those experiencing a more severe course of the earlier infection. Interestingly, it was observed that the deceased patients had higher levels of leukocytes, neutrophils, and markers of inflammations, while the levels of lymphocytes and haemoglobin were significantly lower in relation to the patients who recovered [13]. Computed tomography revealed a higher degree of lung involvement in deceased patients [14]. Some studies indicate that the degree of lung involvement in unvaccinated patients is considerably higher than in those vaccinated with two doses of COVID-19 vaccines [15]. Other diseases that influence the severity of the course of COVID-19 and the risk of death are chronic renal failure, cardiovascular diseases, hypertension, cerebrovascular diseases, COPD, malignant tumours, diabetes, immune deficiency, and obesity [16,17,18]. It should be emphasised that the present study group of unvaccinated hospital patients had a relatively low number of comorbidities, but 62% of the patients had hypertension indicated as a risk factor for death due to COVID-19. Considering the above, it can be stated that the key aspect that influences the mortality of patients is age, which is often associated with multiple comorbidities and a general worsening of health. Studies show that older age and a larger number of comorbidities can explain long hospitalisation times and abnormal blood parameters [3]. 

The literature review clearly indicates that unvaccinated patients have significantly worse clinical parameters [15] and a poorer prognosis [19], and their mortality rate is 5-times higher than that of vaccinated patients (e.g., the mortality rates in a study by Haas EJ et al. were 64.2% in unvaccinated patients and 12.4% in those vaccinated [20]). Some sources indicate that unvaccinated patients have an 11-fold increased risk of death [21]. Despite the fact that deaths due to COVID-19 also occur in vaccinated patients, it can be seen in unvaccinated patients that they appear mainly in elderly people with multiple comorbidities (especially diabetes, acute asthma, liver disease, and renal failure) [13]. Interestingly, in this case, it was also observed that male gender was a predictor of death, despite vaccination status [22,23], which may result from the fact that women are generally characterised by a better immune response to internal and external antigens than men [15,24], and this advantage can also be reflected in the effectiveness of the vaccinations in these two groups [25]. We also underline that higher levels of inflammation markers (CRP, ferritine, D-dimer) correlate with the increased severity of COVID-19 and are associated with worse results in these patients [26]. 

It is significant that this type of study does not usually assess differences in clinical and laboratory parameters between vaccinated and unvaccinated patients [27]. Additionally, studies often do not differentiate between patients vaccinated with one dose and those vaccinated with all possible doses in a given vaccination regime. For example, Sagiraju HKR et al. show that the percentage of fully vaccinated patients in their study was 3% [28,29], and this may have a significant impact on outcomes.

Similar to all studies, this present study has some limitations. First of all, it comprised a relatively small group of patients. The analysis involved all the unvaccinated patients hospitalised at the National Hospital in Warsaw. Due to the significance of this hospital and the lack of a catchment area for hospital services in Poland, the patients were diverse in terms of demographic and clinical characteristics. It should be emphasised that we have not monitored the condition of the patients following hospitalisation and we do not know if the mortality rate was higher in this group. Despite these limitations, to our knowledge, this is the first publication of this type that deals with the course of COVID-19 in unvaccinated patients in Poland.

## 5. Conclusions

This study indicates the relatively severe course of COVID-19 among unvaccinated patients, with high mortality. Our literature review shows that the course of the disease and clinical outcomes in unvaccinated patients are significantly worse than in those vaccinated against COVID-19. For this reason, it is extremely pertinent to undertake continuous efforts to vaccinate the highest percentage of the population possible, including with boosters. These coordinated activities and extensive health education on vaccinations can lead to herd immunity and largely reduce mortality due to COVID-19. Our work indicates that due to the severe clinical effects associated with COVID-19 infection, it is essential to develop and implement strategies to stop the retreat from vaccination, both at the governmental and lower administrative levels. Further work is needed to confirm our results, including mortality due to COVID-19 in the group of unvaccinated patients.

## Figures and Tables

**Table 1 vaccines-11-00675-t001:** Demographic characteristics.

Characteristic	N = 50
**Age [years]**	
Mean (SD)	66.82 (3.02)
Range	60, 73
**Sex**	
Female	25 (50.0%)
Male	25 (50.0%)
**Anthropometric data: Body mass [kg]**	
Median [IQR]	80.0 [74.0, 84.0]
Range	52, 117
**Anthropometric data: Height [cm]**	
Median [IQR]	170.0 [164.0, 176.0]
Range	156, 190
**Anthropometric data: BMI [kg/m^2^]**	
Median [IQR]	27.26 [25.4, 29.05]
Range	21.37, 40.01

**Table 2 vaccines-11-00675-t002:** Data on comorbidities.

Comorbidity	N = 50
**Hypertension**	
Yes	31 (62.0%)
No	19 (38.0%)
**Diabetes**	
Yes	13 (26.0%)
No	37 (74.0%)
**Autoimmune diseases**	
Yes	5 (10.0%)
No	45 (90.0%)
**Atherosclerosis**	
Yes	3 (6.0%)
No	47 (94.0%)
**Asthma/COPD**	
Yes	3 (6.0%)
No	47 (94.0%)
**Nephropathy**	
Yes	3 (6.0%)
No	47 (94.0%)
**Liver diseases**	
Yes	2 (4.0%)
No	48 (96.0%)
**Nicotine use**	
Yes	4 (8.0%)
No	46 (92.0%)

**Table 3 vaccines-11-00675-t003:** Data on treatment outcomes.

Characteristic	N = 50
**Days of hospitalisation**	
Median [IQR]	13.0 [9.0, 18.0]
Range	4, 104
**ICU stay**	
Yes	20 (40.0%)
No	30 (60.0%)
**ICU length of stay [days] (in the group of the patients admitted to the ICU)**	
Median [IQR]	10 (5.5, 17)
Range	1, 35
**In-hospital death**	
Yes	17 (34.0%)
No	33 (66.0%)

**Table 4 vaccines-11-00675-t004:** Data on patient condition on admission.

Characteristic	N = 50
**Heart rate**	
Mean (SD)	88.54 (17.15)
Range	50, 149
**Systolic blood pressure**	
Mean (SD)	135.03 (25.4)
Range	93.0, 184.0
**Diastolic blood pressure**	
Mean (SD)	80.12 (13.42)
Range	42.0, 111.0
**Temperature**	
Median [IQR]	36.2 [36.1, 36.5]
Range	35.6, 38.0
O_2_ **saturation during supplementation**	
Median [IQR]	96.0 [94.0, 97.0]
Range	85.0, 100.0
**Blood pO^2^**	
Median [IQR]	54.1 [38.35, 73.15]
Range	19.1, 139.0
**Blood pH**	
Median [IQR]	7.472 [7.427, 7.486]
Range	7.196, 7.720
**RR**	
Median [IQR]	18.0 [15.0, 20.0]
Range	12.0, 26.0
**Type of oxygen therapy on admission**	
None	18 (36.0%)
Nasal cannula	6 (12.0%)
Simple oxygen mask	17 (34.0%)
Non-rebreather mask	0 (0.0%)
High-flow oxygen therapy	2 (4.0%)
NIV	0 (0.0%)
Respirator therapy	
Median [IQR]	30.0 [20.0, 80.0]
Range	0, 95.0

**Table 5 vaccines-11-00675-t005:** Data on patient condition during hospitalisation.

Characteristic	N = 50
**Type of oxygen therapy on admission**	
None	7 (14.0%)
Nasal cannula	18 (36.0%)
Simple oxygen mask	6 (12.0%)
Non-rebreather mask	17 (34.0%)
High-flow oxygen therapy	0 (0.0%)
NIV	2 (4.0%)
Respirator therapy	0 (0.0%)
**Type of oxygen therapy during hospitalisation**	
None	2 (4.0%)
Nasal cannula	14 (28.0%)
Simple oxygen mask	4 (8.0%)
Non-rebreather mask	8 (16.0%)
High-flow oxygen therapy	3 (6.0%)
NIV	4 (8.0%)
Respirator therapy	15 (30.0%)
**Change of oxygen therapy during hospitalisation**	
No change	15 (30.0%)
Worsening	35 (70.0%)
Improvement	0 (0%)

**Table 6 vaccines-11-00675-t006:** Data on medications administered during hospitalisation.

Characteristic	N = 50 (%)
**Dexamethasone**	
Yes	46 (92.0%)
No	4 (8.0%)
**Low-molecule Heparin**	
Yes	49 (98.0%)
No	1 (2.0%)
**Baricitinib**	
Yes	10 (20.0%)
No	40 (80.0%)
**Remdesivir**	
Yes	8 (16.0%)
No	42 (84.0%)

**Table 7 vaccines-11-00675-t007:** Severity of COVID-19 vs. dexamethasone medication, comparison between groups.

	Medications Used during Hospitalisation: Dexamethasone/Demezon			
Characteristic	Overall, N = 50	Yes, N = 46	No, N = 4	*p*-Value ^1^
**Days of hospitalisation**				0.145
Median [IQR]	13.0 [9.0, 18.0]	13 [9.0, 17.0]	8.5 [5.5, 12.0]	
Range	4, 104	6, 46	4, 14	
**ICU stay**				0.523
Yes	20 (40.0%)	19 (41.30%)	0 (0.0%)	
No	30 (60.0%)	27 (58.70%)	10 (100.0%)	
**ICU length of stay [days] (in the group of the** **patients admitted to the ICU (N = 20)**				0.490
Median [IQR]	10 (5.5, 17)	10 [5.0, 18.0]	6 [6.6]	
Range	1, 35	1, 35	6, 6	
**Imaging examination: CT%**				0.249
Median [IQR]	30.0 [20.0, 80.0]	32.5 [25.0, 80.0]	15.5 [0.5, 55.0]	
Range	0, 95.0	1, 95	0, 80	
**Type of oxygen therapy on admission**				0.29
None	7 (14.0%)	5 (10.87%)	2 (20.0%)	
Nasal cannula	18 (36.0%)	17 (36.96%)	1 (25.0%)	
Simple oxygen mask	6 (12.0%)	6 (13.04%)	0 (0%)	
Non-rebreather mask	17 (34.0%)	16 (34.78%)	1 (25.0%)	
High-flow oxygen therapy	0 (0.0%)	0 (0.0%)	0 (0.0%)	
NIV	2 (4.0%)	2 (4.35%)	0 (0.0%)	
Respirator therapy	0 (0.0%)	0 (0.0%)	0 (0.0%)	
**Type of oxygen therapy during hospitalisation**				**<0.001**
None	2 (4.0%)	0 (0.0%)	2 (20.0%)	
Nasal cannula	14 (28.0%)	13 (28.26%)	1 (25.0%)	
Simple oxygen mask	4 (8.0%)	4 (8.70%)	0 (0%)	
Non-rebreather mask	8 (16.0%)	8 (17.39%)	0 (0%)	
High-flow oxygen therapy	3 (6.0%)	3 (6.52%)	0 (0.0%)	
NIV	4 (8.0%)	4 (8.70%)	0 (0.0%)	
Respirator therapy	15 (30.0%)	14 (30.43%)	1 (25.0%)	
**Change of oxygen therapy during hospitalisation**				**0.041**
No change	15 (30.0%)	12 (26.09%)	3 (75.0%)	
Worsening	35 (70.0%)	34 (73.91%)	1 (25.0%)	
Improvement	0 (0%)	0 (0%)	0 (0%)	
**In-hospital death**				0.692
Yes	17 (34.0%)	16 (34.78%)	1 (25.0%)	
No	33 (66.0%)	30 (65.22%)	3 (75.0%)	

^1^ Chi^2^ Pearson, *t*-test.

**Table 8 vaccines-11-00675-t008:** Severity of COVID-19 vs. heparin medication, comparison between groups.

	Medications Used during Hospitalisation: Heparin (Clexane, Fragmin, Neoparin)			
**Characteristic**	**Overall, N = 50**	**Yes, N = 49**	**No, N = 1**	***p*-Value ^1^**
**Days of hospitalisation**				0.805
Median [IQR]	13.0 [9.0, 18.0]	13 [9.0, 17.0]	8.0 [8.0, 8.0]	
Range	4, 104	4, 46	8, 8	
**ICU stay**				0.409
Yes	20 (40.0%)	20 (40.82%)	0 (0%)	
No	30 (60,0%)	29 (59.18%)	1 (100%)	
**ICU length of stay [days] (in the group of the patients admitted to the ICU)**				
Median [IQR]	10 (5.5, 17)	10 [5.5, 17.0]	0 [0, 0]	
Range	1, 35	1, 35	0, 0	
**Imaging examinations: CT %**				0.630
Median [IQR]	30.0 [20.0, 80.0]	30,0 [20.0, 80.0]	30.0 [30.0, 30.0]	
Range	0, 95.0	0, 95	30, 30	
**Type of oxygen therapy on admission**				0.770
None	7 (14.0%)	7 (14.29%)	0 (0%)	
Nasal cannula	18 (36.0%)	17 (34.69%)	1 (100%)	
Simple oxygen mask	6 (12.0%)	6 (12.24%)	0 (0.0%)	
Non-rebreather mask	17 (34.0%)	17 (34.69%)	0 (0.0%)	
High-flow oxygen therapy	0 (0.0%)	0 (0.0%)	0 (0.0%)	
NIV	2 (4.0%)	2 (4.08%)	0 (0.0%)	
Respirator therapy	0 (0.0%)	0 (0.0%)	0 (0.0%)	
**Type of oxygen therapy during hospitalisation**				0.854
None	2 (4.0%)	2 (4.08%)	0 (0%)	
Nasal cannula	14 (28.0%)	13 (26.53%)	1 (100%)	
Simple oxygen mask	4 (8.0%)	4 (8.16%)	0 (0.0%)	
Non-rebreather mask	8 (16.0%)	8 (16.33%)	0 (0.0%)	
High-flow oxygen therapy	3 (6.0%)	3 (6.12%)	0 (0.0%)	
NIV	4 (8.0%)	4 (8.16%)	0 (0.0%)	
Respirator therapy	15 (30.0%)	15 (30.61%)	0 (0.0%)	
**Change of oxygen therapy during hospitalisation**				0.122
No change	15 (30.0%)	14 (28.57%)	1 (100%)	
Worsening	35 (70.0%)	35 (71.43%)	0 (0.0%)	
Improvement	0 (0%)	0 (0%)	0 (0.0%)	
**In-hospital death**				0.468
Yes	17 (34.0%)	17 (34.69%)	0 (0%)	
No	33 (66.0%)	32 (65.31%)	1 (100%)	

^1^ Chi^2^ Pearson, *t*-test.

**Table 9 vaccines-11-00675-t009:** Severity of COVID-19 vs. baricitinib medication, comparison between groups.

	Medications Used during Hospitalisation: Baricitinib			
Characteristic	Overall, N = 50	Yes, N = 10	No, N = 40	*p*-Value ^1^
**Days of hospitalisation**				**0.011**
Median [IQR]	13.0 [9.0, 18.0]	16.0 [11.0, 24.0]	11.5 [8.5, 16.0]	
Range	4, 104	9, 46	4, 34	
**ICU stay**				0.149
Yes	20 (40.0%)	6 (60.0%)	14 (35.0%)	
No	30 (60.0%)	4 (40.0%)	26 (65.0%)	
**ICU length of stay [days] (in the group of** **patients admitted to the ICU)**				0.629
Mean (SD)	10 (5.5, 17)	10.0 (5.0, 15.0)	10.5 (6.0 18.0)	
Range	1, 35	4, 35	1, 22	
**Imaging examinations: CT %**				**0.026**
Median [IQR]	30.0 [20.0, 80.0]	70.0 [40.0, 80.0]	30.0 [15.0, 75.0]	
Range	0, 95.0	30, 95	0, 95	
**Type of oxygen therapy on admission**				**0.04**
None	7 (14.0%)	0 (0.0%)	7 (17.5%)	
Nasal cannula	18 (36.0%)	1 (10.0%)	17 (42.5%)	
Simple oxygen mask	6 (12.0%)	2 (20.0%)	4 (10.0%)	
Non-rebreather mask	17 (34.0%)	7 (70.0%)	10 (25.0%)	
High-flow oxygen therapy	0 (0.0%)	0 (0.0%)	0 (0%)	
NIV	2 (4.0%)	0 (0.0%)	2 (5.0%)	
Respirator therapy	0 (0.0%)	0 (0.0%)	0 (0.0%)	
**Type of oxygen therapy during hospitalisation**				**0.005**
None	2 (4.0%)	0 (0.0%)	2 (5.0%)	
Nasal cannula	14 (28.0%)	0 (0.0%)	14 (35.0%)	
Simple oxygen mask	4 (8.0%)	1 (10.0%)	3 (7.5%)	
Non-rebreather mask	8 (16.0%)	1 (10.0%)	7 (17.5%)	
High-flow oxygen therapy	3 (6.0%)	3 (30.0%)	0 (0%)	
NIV	4 (8.0%)	2 (20.0%)	2 (5.0%)	
Respirator therapy	15 (30.0%)	3 (30.0%)	12 (30.0%)	
**Change of oxygen therapy during hospitalisation**				**0.021**
No change	15 (30.0%)	0 (0%)	15 (37.5%)	
Worsening	35 (70.0%)	10 (100.0%)	25 (62.5%)	
Improvement	0 (0%)	0 (0%)	0 (0%)	
**In-hospital death**				0.654
Yes	17 (34.0%)	4 (40.0%)	13 (32.5%)	
No	33 (66.0%)	6 (60.0%)	27 (67.5%)	

^1^ Chi^2^ Pearson, *t*-test.

**Table 10 vaccines-11-00675-t010:** Severity of COVID-19 vs. remdesivir medication, comparison between groups.

	Medications Used during Hospitalisation: Remdesivir (Veklury)			
Characteristic	Overall, N = 50	Yes, N = 8	No, N = 42	*p*-Value ^1^
**Days of hospitalisation**				0.653
Median [IQR]	13.0 [9.0, 18.0]	12.5 [8.0, 20.0]	12.5 [9.0, 17.0]	
Range	4, 104	7, 34	4, 46	
**ICU stay**				0.345
Yes	20 (40.0%)	2 (25.0%)	18 (42.86%)	
No	30 (60.0%)	6 (75.0%)	24 (57.14%)	
**Length of ICU stay [days] (N = 20)**				0.632
Median [IQR]	10 (5.5, 17)	14.5 [7,0,0 22.0]	10.0 [5.0, 16.0]	
Range	1, 35	7, 22	1, 35	
**Imaging examinations: CT %**				0.743
Median [IQR]	30.0 [20.0, 80.0]	32.5 [20.0, 65.0]	30.0 [20.0, 80.0]	
Range	0, 95.0	1, 95	0, 95	
**Type of oxygen therapy on admission**				0.470
None	7 (14.0%)	1 (12.5%)	6 (14.29%)	
Nasal cannula	18 (36.0%)	5 (62.5%)	13 (30.95%)	
Simple oxygen mask	6 (12.0%)	1 (12.5%)	5 (11.90%)	
Non-rebreather mask	17 (34.0%)	1 (12.5%)	16 (38.10%)	
High-flow oxygen therapy	0 (0.0%)	0 (0.0%)	0 (0.0%)	
NIV	2 (4.0%)	0 (0.0%)	2 (4.76%)	
Respirator therapy	0 (0.0%)	0 (0.0%)	0 (0.0%)	
**Type of oxygen therapy during hospitalisation**				0.210
None	2 (4.0%)	0 (0.0%)	2 (4.76%)	
Nasal cannula	14 (28.0%)	4 (50.0%)	10 (23.81%)	
Simple oxygen mask	4 (8.0%)	2 (25.0%)	2 (4.76%)	
Non-rebreather mask	8 (16.0%)	0 (0%)	8 (19.05%)	
High-flow oxygen therapy	3 (6.0%)	0 (0%)	3 (7.14%)	
NIV	4 (8.0%)	0 (0%)	4 (9.52%)	
Respirator therapy	15 (30.0%)	2 (25.0%)	13 (30.95%)	
**Change of oxygen therapy during hospitalisation**				**0.029**
No change	15 (30.0%)	5 (62.5%)	10 (23.81%)	
Worsening	35 (70.0%)	3 (37.5%)	32 (76.19%)	
Improvement	0 (0%)	0 (0%)	0 (0%)	
**In-hospital death**				0.161
Yes	17 (34.0%)	1 (12.5%)	16 (38.10%)	
No	33 (66.0%)	7 (87.5%)	26 (61.90%)	

^1^ Chi^2^ Pearson, *t*-test.

## Data Availability

All data are available from corresponding author.

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
