# Peer review of "The Course of COVID-19 among Unvaccinated Patients—Data from the National Hospital in Warsaw, Poland"

_vaccines, 2023, doi:10.3390/vaccines11030675_

Round 1

Reviewer 1 Report

1.     The authors are advised to check the grammatical errors throughout the manuscript.

2.     There are a few longer sentences, do split the lengthy sentence so that it would be easy for the readers to understand correctly.

3.     It is advised to outline the article’s structure towards the end of the introduction. The entire manuscript needs to be revised, and a proper English language check should be made.

4.     The materials ad methods section can be a little more descriptive.

5.     This work lacks proper connection among the subsections. More details and recent case studies are required to support the text.

6.     The discussion section needs to be carefully revised. Please ensure your discussion section underscores the scientific value-added of your paper and the applicability of your findings/results. Please revise your discussion part in more detail. It would be best if you enhanced your contributions and limitations, underscored the scientific value-added of your paper, and/or the applicability of your findings/results and future study in this session.

7.     The conclusion section does not correctly address the important points and future perspectives. The author should add a few important points.

Reviewer 2 Report

ABSTRACT

"However, in Poland, 40% of the population remain unvaccinated."

add month and year so readers know when this statistic applies

"Despite the existence of many studies showing the effects of COVID-19 or the impact of vaccinations on the course of the disease, relatively few publications detail the course of COVID-19 solely in unvaccinated hospital patients"

delete this sentence. I am not sure it is true, and it doesn't add information about your paper

"The basic objective of this study then was to describe the natural history COVID-19 in unvaccinated hospital patients in Warsaw, Poland ."

wordy. just write

"The objective of this study was to describe the natural history COVID-19 in unvaccinated hospital patients in Warsaw, Poland.

"and 34% subsequently died."

rewrite as

"and 34% subsequently died prior to the end of the study".

or whatever end date is appropriate.

eventually everyone will die...

INTRODUCTION

"The full COVID-19 vaccination schedule for Pfizer and Moderna vaccines is two doses. The Johnson & Johnson vaccination is a single dose. The others are boosters [2]."

I don't think this is true anymore. Maybe just delete these sentences. They are not terribly relevant to your paper, and it is easy for researchers to look up current recommended guidelines.

"The current study takes into account patients hospitalized at the main covid hospital in Poland - the national hospital, and is the only study of this type."

just write

"The current study takes into account patients hospitalized at the main covid hospital in Poland - the national hospital."

claims of novelty don't belong in original research papers

there are a couple of other sentences where you talk about first study (first and last in Discussion) - please delete these as well

Maethods

"The analysis comprised 50 patients hospitalised during the study period who had not previously been vaccinated against COVID-19."

explain why 50?

why not ALL such hospitalized pts?

If not all, how were 50 chosen?

Could the selection have been biased?

Table 2

footnote 1 is missing

Results

Please read a basic text on significant figures (or get advice from a math or science professional). Only report significant figures.

Consider "The average heart rate was 85.54±17.15 bpm, the median systolic blood pressure was 135.03±25.4"

Rewrite as 

"The average heart rate was 86±17 bpm"

etc. as appropriate throughout the manuscript

Ethics Statements

Needs documentation of IRB approval and patient consent.

MAJOR POINT

Please add a paragraph to Discussion (perhaps also Results) about what strains and variants of SARS-CoV-2 were prevalent in Poland during the time of this study. And if they changed over time. Did you get sequence from any of your patients? And discuss if strain affects mortality or could affect mortality.

please be consistent with capitalization (e..g COVID not covid)

Reviewer 3 Report

The authors present a fairly detailed clinical account of the experience of patients admitted to a hospital in Poland and requiring management of their COVID infection. It presents a logical, methodical and clinically accurate description of the disease progression and recovery of unvaccinated patients drawn from a large catchment area of hospital services in Poland.

The study follows the experience of 50 unvaccinated patients and a fairly detailed clinical picture of the course of treatment is provided. Poland unfortunately has a large percentage of COVID unvaccinated citizens.

In spite of up-to-date treatment to manage their COVID infections unfortunately the death rate was 34% in a cohort that was not elderly (age range 60 to 73). The cohort consisted of 25 F and 25 M with comorbidity generally low except for hypertension and diabetes.

The study clearly demonstrates the importance of extending vaccination coverage within a country to enable protection of vulnerable section of the community.  In this case the relatively small dataset nonetheless demonstrated that unvaccinated patients, mostly with co-morbidities, experienced life-threatening effects from their battle with COVID even though they were not even in the highest age bracket of over 75 years. The treatment benefits of Remdesivir are again shown and the sometimes questionable benefit of Dexamethasone in treating COVID with significant respiratory involvement.

The introduction is comprehensive and sound and the methodology is generally clearly stated. The statistical treatment of data appears appropriate and carefully considered, and the conclusions drawn and elaborated upon in the discussion are  reasonable and accurate.

Generally quite a nice paper that focusses on experiences with treating COVID infected unvaccinated patients in Poland and once again demonstrating the very sad outcomes that quite often occur. Overcoming vaccine resistance remains a problem in Poland and in many other countries!

Specific suggestions for improvement

Check throughout MS for accepted spelling of cannula – both text and tables

Define Non-invasive ventilation (NIV) early in the text. Currently it first appears in the text following Table 3

Results

The effects of treatments during hospitalisation

…and were more likely to experience a worsening of their condition (p=0.041). ie delete from this sentence a repeat of “and require a more invasive form of oxygen therapy”

Table 8 legend   … not possible to demonstrate a statistically significant difference.

Table 8 Median IQR check for consistency eg 30.0 [20.0, 80.0] and 30.0 [30.0, 30.0]

…worsening of their condition  and require more invasive forms of oxygen therapy….

Check and correct entry for Table 10, Median IQR entered as 14.5 [7,0,0 22.0] and should be, I believe, 14.5 [7.0, 22.0]

also above under No,  for consistency Median IQR 12.5 [9.0, 17.0]

Discussion

Ist para …the data were collected

2nd para  markers of inflammation…

Para 2, L5…and markers of inflammation, while ….